

# Development of an incoherent broadband cavity enhanced absorption spectrometer for in situ measurements of HONO and NO₂ in China

Jun Duan[1], Min Qin[1*], Bin Ouyang[2], Wu Fang[1], Xin Li[3], Keding Lu[3], Ke Tang[1], Shuaixi Liang[1], Fanhao Meng[1], Zhaokun Hu[1], Pinhua Xie[1,4,5], and Wenqing Liu[1,4,5]

[1]Key Laboratory of Environment Optics and Technology, Anhui Institute of Optics and Fine Mechanics, Chinese Academy of Sciences, Hefei, 230031, China
[2]Department of Chemistry, University of Cambridge, Cambridge, CB2 1EW, UK
[3]College of Environmental Sciences and Engineering, Peking University, Beijing, 100871, China
[4]CAS Center for Excellence in Urban Atmospheric Environment, Institute of Urban Environment, Chinese Academy of Sciences, Xiamen, 361021, China
[5]School of Environmental Science and Optoeclectronic Technology, University of Science and Technology of China, Hefei, 230027, China

*Correspondence to*: Min Qin (mqin@aiofm.ac.cn)

**Abstract.** Gaseous nitrous acid (HONO) is an important source of OH radical in the troposphere. However, its source, especially that during daytime hours remains unclear. We report hereby a home-built instrument for simultaneous unambiguous measurements of HONO and NO₂ with high time resolution based on incoherent broadband cavity enhanced absorption spectroscopy (IBBCEAS). To achieve robust performance and system stability under different environment conditions, the current IBBCEAS instrument has made significant improvements in terms of efficient sampling as well as resistance against vibration and temperature change, and the IBBCEAS instrument also has low-power consumption and a compact design that can be easily deployed on different platforms powered by a high-capacity lithium ion battery. The effective cavity length of the IBBCEAS was determined using the absorption of O₂-O₂ to account for the "shortening" effect caused by the mirror purge flows. The wall loss for HONO was estimated to be 2 % via a HONO standard generator. Measurement precisions (2σ) for HONO and NO₂ are about 180 ppt and 340 ppt in 30 s, respectively. A field inter-comparison was carried out at a rural suburban site in Wangdu, Hebei Province, China. The concentrations of HONO and NO₂ measured by IBBCEAS were compared with a long optical path absorption photometer (LOPAP) and a NOx analyzer (Thermo Electron Model 42i), and the results showed very good agreement, with correlation coefficients (R²) of HONO and NO₂ being ~0.89 and ~0.95, respectively, in addition, vehicle deployments were also tried to enable mobile measurements of HONO and NO₂, demonstrating the promising potential of using IBBCEAS for in situ, sensitive, accurate and fast simultaneous measurements of HONO and NO₂ in the future.



# 1 Introduction

Nitrous acid (HONO) is an important photochemical precursor of hydroxyl radical (OH) through reaction (R1) and has been found worldwide to play a key role in enhancing the formation of photochemical smog and other secondary pollutants such as $O_3$ in polluted atmospheric boundary layers(Pitts et al., 1984;Perner and Platt, 1979;Kleffmann et al., 1998;Kleffmann et al., 2005).

$$HONO + h\nu \rightarrow OH + NO \quad (R1)$$

Recent field and modeling studies have shown that reaction R1 contributes significantly to OH radical budgets not only during the early morning when the other $HO_x$ sources such as the photolysis of $O_3$ and HCHO were still weak, but also throughout the entire day (Alicke et al., 2002;Kleffmann, 2007).

HONO can be emitted directly into the atmosphere through combustion processes such as biomass burning, industrial burning, domestic heating and internal combustion engine burning (Jr et al., 1984;Kurtenbach et al., 2001;Xu et al., 2015). And HONO can also be formed through several homogeneous gas-phase reactions (Pagsberg et al., 1997;Zhang and Tao, 2010;Li et al., 2014) as well as the photolysis of different gaseous nitrophenols (Bejan et al., 2006). In addition to the gas-phase formation pathways, heterogeneous reactions between $NO_2$ and water on surfaces are well known to produce HONO effectively (Kleffmann et al., 1998;Kleffmann et al., 1999). In most occasions, emission, homogeneous gas-phase and heterogeneous reactions were unable to fully match the observed high daytime HONO levels. Thus, photochemical processes were proposed to explain the missing HONO sources. The photolysis of surface-adsorbed nitrate ($NO_3^-$) or nitric acid ($HNO_3$) can lead to $NO_2$ or HONO which then partition to the gas phase (Zhou et al., 2011). Furthermore, Su et al. showed that nitrites could be a source of HONO through soil biological processes (Su et al., 2011). To summarize, many HONO formation mechanisms have been proposed based on laboratory or field studies (Kleffmann, 2007;Spataro and Ianniello, 2014), and we suspect that the relative importance of these mechanisms varies depending on a number of factors such as geographic location (rural vs. urban), surface properties at the interested site, time of the day as well (high vs. low actinic flux) as meteorological factors such as relative humidity.

Development of analytical instruments that are capable of providing accurate and fast HONO measurements is one of the primary and most crucial steps towards a full understanding of the atmospheric behaviours of HONO. So far three major categories of techniques have been developed for HONO field measurements: wet chemistry, mass spectrometry and optical spectroscopy. Wet chemical techniques mainly contain Annular Denuder-ion chromatography (IC) (Ferm and Sjödin, 1985), Wet Denuder-IC (Neftel et al., 1996), Mist Chamber-IC (Dibb et al., 2004), DNPH(2,4-dinitrophenylhydrazone)-HPLC (Zhou et al., 1999), and long optical path absorption photometer (LOPAP) (J et al., 2001). These offer high sensitivities with detection limits in the lower ppt, but they generally measure gaseous HONO by absorbing it with liquid solutions to convert to nitrite anion. As a result, they are indirect measurements and may potentially suffer from chemical interferences and sampling artifacts. More recently, chemical ionization mass spectrometry has been used to measure HONO using $I^-$



ionization (Veres et al., 2015). Reported $3\sigma$ detection limit was ~30 ppt, but dynamic calibration was required to derive sensitivity coefficient towards HONO detection.

Spectroscopic techniques for HONO detection are mostly based on differential optical absorption spectroscopy (DOAS) (Perner and Platt, 1979), incoherent broadband cavity enhanced absorption spectroscopy (IBBCEAS) (Gherman et al., 2008),

tunable diode laser spectroscopy (TDLS) (Li et al., 2008) or laser induced fluorescence (LIF) (Liao et al., 2006).

Among these, IBBCEAS was first introduced by Fiedler et al (Fiedler et al., 2003). It offers in situ, high-sensitivity, chemical interference-free and direct concentration measurements (like DOAS) while maintaining high spatial resolution. IBBCEAS has been successfully deployed to measure many atmospheric trace gases like $NO_2$, HONO, IO, $O_3$, $I_2$, IO, OIO, $SO_2$, $NO_3$, $N_2O_5$, CHOCHO, methylglyoxal and aerosol extinction coefficients. IBBCEAS has gradually been applied to the

study of HONO (Gherman et al., 2008;Wu et al., 2011;Wu et al., 2014;Reeser et al., 2013;Scharko et al., 2014;Min et al., 2016),with continued improvements in sensitivity.

But the results of the field inter-comparison between IBBCEAS and LOPAP in Hong Kong were far from satisfactory, and there were some inconsistencies in the quantitative assessment of HONO, and Wu et al indicated that IBBCEAS system stability might be perturbed by environmental and experimental conditions (such as variation of temperature, pressure,

humidity, environmental vibration, etc.), and the accurate retrieval of HONO concentration was very challenging (Wu et al., 2014).

In this paper, we describe a home-built incoherent broadband cavity enhanced absorption spectrometer in detail. To achieve robust performance and system stability under different environment conditions, the current IBBCEAS instrument has made significant improvements in terms of efficient sampling as well as resistance against vibration and temperature change. The

effective cavity length of the IBBCEAS, the wall loss of HONO, and the Allan variance analysis are discussed. We deployed the IBBCEAS instrument to simultaneously measure ambient concentrations of HONO and $NO_2$ at a rural site (Wangdu, Hebei Province) in Northern China from June 26[th] to July 9[th], 2014, and compared the results with a LOPAP instrument for HONO measurements and a $NO_x$ analyzer for $NO_2$ measurements. In December 2014, the IBBCEAS instrument was successfully deployed in a car for mobile measurements of HONO and $NO_2$ and results from this deployment will also be

briefly presented.

## 2 IBBCEAS Instrument description

### 2.1 Theory of IBBCEAS

IBBCEAS is based on the well-known Beer-Lambert law. The optical extinction $\alpha\left(\lambda\right)$ is related to the reflectivity of mirrors along and the wavelength. The light intensities transmitted through an empty cuvette and sample-filled cuvette are

measured along with the effective absorption path length. Optical extinction $\alpha\left(\lambda\right)$ can be described as



$$\alpha(\lambda) = \left[ \frac{1-R(\lambda)}{d_{eff}} + \alpha_{Ray}(\lambda) \right] \cdot \frac{I_0(\lambda) - I(\lambda)}{I(\lambda)} \quad (1)$$

where $R(\lambda)$ is the wavelength-dependent reflectivity of mirrors, $I(\lambda)$ are the light intensities transmitted through the optical cavity with the absorbing species inside, $I_0(\lambda)$ are the light intensities transmitted through the empty optical cavity without absorbing trace gases or aerosols at the same temperature and pressure as the sample spectrum, $\alpha_{Ray}(\lambda)$ is the wavelength-dependent extinction coefficient for Rayleigh scattering at the same pressure, and $d_{eff}$ is the effective cavity length. Concentrations of the light-absorbing gases are obtained by applying nonlinear least squares fittings to the observed absorption coefficients using the DOASIS program (Kraus, 2006).

**2.2 Optical system**

The IBBCEAS instrument consists of an UV-LED light source, two highly reflective mirrors, an optical cavity, and a spectrometer (Fig. 1). The light emitted by the UV-LED (365 nm, LZ1-00UV00, LEDengin) was directly coupled to the optical cavity by an achromatic lens (f = 60 mm, Edmund), which was then reflected back and forth between the two highly reflective mirrors (CRD Optics) resulting in a long photon residence time and long optical path length. The light transmitted through the cavity was coupled into an optical fibre (600 μm, Ocean Optics) by another achromatic lens ($f$ = 60 mm, Edmund) which was finally received by a CCD spectrometer (QE65000, Ocean Optics).

The near-ultraviolet LED was supplied by a 1000 mA constant current source and emitted about 1680 mW optical power at around 365 nm with full width at half maximum (FWHM) of 13 nm. To maintain the stability of the light source, the LED was mounted on a temperature-controlled cooling stage stabilized at 20 ± 0.05 °C by a temperature sensor (PT1000) and a thermoelectric cooler. The highly reflective mirrors (1 m radius of curvature) were mounted on each end of the PFA optical cavity (separated by 55 cm). While spectral filtering of the LED's FWHM seems unnecessary, there is unwanted stray light caused by slight impurity in the chip epitaxy at wavelengths outside the high reflectivity range. Thus, a band-pass filter (BG3, Newport) was placed at the end of the optical cavity to prevent it from entering the spectrometer.

**2.3 Flow system**

Strong Mie scattering during high aerosol loading periods greatly shortens the effective optical length and degrades the IBBCEAS detection sensitivity (Wu et al., 2014). As a result, aerosol particles were removed by a 1 μm PTFE filter membrane (Tisch Scientific) placed in front of our IBBCEAS instrument inlet. Further protection was done by continuously purging the two mirrors with high purity $N_2$ to block the contact between the mirrors and the sample airflow. Flow rates of these two purge lines were controlled at ~0.1 SLPM (Standard Litres per Minute) by two mass flow controllers (MFCs, CS200A, Sevenstar). A third MFC was used to control the sample flow rate at 1 ~ 3 SLPM.





For the measurements in rural site or farmland, relative humidity in sampling flow was usually very high, especially in the mornings. To avoid water condensation in the inlet line, sampling tube of our IBBCEAS instrument was heated to using temperature-controlled heating wires.

To minimize the wall loss of HONO, materials for the inlet tube and the optical cavity were exclusively PFA Teflon, well
known for its chemical inertness. As shown in Fig. 1(a), ambient air was drawn through the particle filter and the air inlet tube (OD 4 mm, length 3 m) by a gas pump (KNF) at 6 SLPM so that the residence time of the sampled air in the inlet tube was less than 0.5 s. Part of the air sample went into the optical cavity, and the rest went through a bypass tube. A rotameter was used to control the air flow rate in the bypass gas line. Air pressure, temperature and humidity were measured at the outlet of the cavity.

**2.4 Hardware**

The current IBBCEAS instrument has seen significant improvements over vibration and temperature resistance and power consumption to achieve robust performance in different field environments. For example, we found that the parameters of the CCD spectrometer changed with temperature especially when it dropped below 5 °C. To ensure the IBBCEAS can work across a wide ambient temperature range, insulation foam was attached to the internal walls of the instrument shell, and
infrared heating lamps and cooling fans were incorporated in the IBBCEAS instrument so that the internal temperature of the instrument can be regulated to suitable temperatures using a PID feedback loop.

To overcome the effect of vibrations for the measurements on mobile platforms, the PFA optical cavity were enfolded in the tailor-made stainless steel pipe shroud and fixed on a 12-mm thick aluminium plate, and anti-vibration mounts were utilized at the bottom of the IBBCEAS instrument frame. Experimental results have demonstrated that normal vibrations would not
affect spectral stability of this instrument.

The IBBCEAS instrument has low-power consumption. The UV-LED light source, the temperature control system of UV-LED, and the gas pump are rated at 5 W, 15 W and 35 W, respectively. The total power consumption of the spectrometer, the various sensors, mass flowmeters and the thermostat cooling fan is about 40 W. Thus, the total power usage of the instrument is less than 100 W excluding the notebook computer. It can be powered for dozens of hours by a high capacity
lithium ion battery.

A photograph of the home-built IBBCEAS instrument used in rural measurements (CAREBEIJING-NCP 2014 campaign) is shown in Fig. 1(b), and the whole system is controlled via a computer. The photograph shows that the IBBCEAS instrument is smaller (1100 mm (L) × 220 mm (W) × 270 mm (H)) than other IBBCEAS instruments (Kennedy et al., 2011;Wu et al., 2014;Min et al., 2016). It can be easily placed in a car or airplane for field applications to measure HONO and $NO_2$.




## 3 Data analysis

### 3.1 Calibrations of Mirror Reflectivity

IBBCEAS is different from differential optical absorption spectroscopy (DOAS). The optical path length in DOAS can be measured directly, but according to Eq. 1, the mirror reflectivity in IBBCEAS must be accurately determined before

concentration calculation. Here, methods widely used to calibrate mirror reflectivity (through the difference of Rayleigh scattering or a known concentration of an absorber) were compared.

The first method uses gases (He and $N_2$) with different Rayleigh scattering cross sections. The mirror reflectivity curve can be calculated with Eq. 2 as described by Washenfelder et al(Washenfelder et al., 2008).

$$R(\lambda) = 1 - d_0 \cdot \frac{\frac{I_{N_2}(\lambda)}{I_{He}(\lambda)} \cdot \alpha_{Ray}^{N_2}(\lambda) - \alpha_{Ray}^{He}(\lambda)}{1 - \frac{I_{N_2}(\lambda)}{I_{He}(\lambda)}} \qquad (2)$$

where $I_{N_2}(\lambda)$ and $I_{He}(\lambda)$ are the cavities throughputs when the cavity is filled with $N_2$ or He, respectively, $\alpha_{Ray}^{N_2}(\lambda)$ and $\alpha_{Ray}^{He}(\lambda)$ are the extinction coefficients caused by Rayleigh scattering of $N_2$ and He, respectively, and $d_0$ is the length of the cavity. Note that $d_0$ is different from $d_{eff}$. Because the entire cavity is filled with the calibration gases in this step. $N_2$ (99.9999%) or He (99.9999%) are added to the cavity at 0.1 SLPM. The IBBCEAS instrument then records stable $N_2$ and He spectra and the reflectivity of the mirrors can be calculated from Eq. 2.

The second method uses a known concentration of $NO_2$. To get the accurate concentration of $NO_2$, we used a mixture of standard NO and $O_3$ and then diluted it in high-purity $N_2$ gas, and mirror reflectivity curve at the same temperature and pressure as the normal sampling measurements can be calculated with the reverse process of Eq. 1.

During CAREBEIJING-NCP campaign, we calibrated the mirror reflectivity by these two different methods in a container and the results are shown in Fig. 2. Reflectivity curves obtained from the two methods between 359 – 380 nm are shown in

Fig. 2(c), which exhibit excellent agreement. The mirror reflectivity at HONO peak absorption wavelength (368.2 nm) was found to be ~0.99983 during CAREBEIJING-NCP campaign.

### 3.2 Effective cavity length

Reflectivity of the mirrors employed in IBBCEAS is usually > 99.95%. If contaminants in the airflow come into contact with the mirrors for a long time, aerosol particles or organic species might get adsorbed on the surfaces of the mirrors, causing

reflectivity degradation. Without protection, surfaces of highly reflective mirrors are easily contaminated particularly in severely polluted areas like northern China. A common practice, as is also adopted in the current study, is that two purge lines are placed in front of the mirrors, creating a volume that is filled with high-purity $N_2$ instead of sample gas. This has the effect of reducing the effective absorption pathlength inside the cavity which should be characterised through proper experiments.



### 3.2.1 Past practices on determining the effective cavity length

To quantify this effect, Meinen et al. (Meinen et al., 2010) utilized computational fluid dynamics to calculate the characteristics of the flow inside the cavity. Thalman and Volkamer (Thalman and Volkamer, 2010) and Kennedy et al. (Kennedy et al., 2011) determined the effective cavity length through calibrations with known amounts of water vapour.

Hoch et al. (Hoch et al., 2014) placed the gas inlet of the IBBCEAS instrument in the middle of cavity, and the effective cavity length was taken as being the distance between the two gas outlets. Similarly, Vaughan et al. estimated the effective absorption path length as the distance between the point of introduction of the sample and the pump outlet (Vaughan et al., 2008). Johansson et al. believed that the effective cavity length is a function of the purge flow rate for a certain sample flow. They calibrated $(1 - R)/ d_{\text{eff}}$ using known concentrations of $I_2$ (Johansson et al., 2014). However, Reeser et al. (Reeser et al.,

2013) and Scharko et al. (Scharko et al., 2014) reckoned that the effect of the mirrors' purge flow on the final measured concentration was negligible.

### 3.2.2 Determining the effective cavity length by pure oxygen

In this paper, pure oxygen was utilized to quantify the effective cavity length making use of the fact that the $O_2$-$O_2$ collision pair has two absorption peaks at near 361 nm and 380 nm. Furthermore, it is easy to obtain, and its concentration can be

known with high accuracy.

Particle-free standard gases were used in this experiment. Nominal $[O_2]$ can be retrieved from the observed absorption of the $O_2$-$O_2$ pair, with or without purging flows. The value of $d_{\text{eff}}$ is calculated as:

$$d_{\text{eff}} = d_0 \times \frac{[O_2]_{\text{on}}}{[O_2]_{\text{off}}} \quad (3)$$

where $[O_2]_{\text{on}}$ and $[O_2]_{\text{off}}$ are the retrieved $[O_2]$ (from the $O_2$-$O_2$ absorption) with or without the purge flows. Using this

method, the effective cavity length of the IBBCEAS instrument was determined to be $48.0 \pm 0.5$ cm when the sample flow of pure oxygen is 1 SLPM and the two purge flows are both 0.1 SLPM, as shown in Fig. 3.

The above result shows that in our case the effect of the mirrors' purge flow on the final measured concentration is not negligible. We also note that for the current IBBCEAS instrument, the determined effective absorption path length is not equal to the distance between the inlet and outlet of the sample flow.

### 3.2.3 Verification of the effective cavity length

To verify the accuracy of the effective cavity length derived using pure oxygen, results from this 365 nm IBBCEAS instrument were compared with those from another 460 nm LED IBBCEAS instrument that runs without purge flows. To distinguish the two IBBCEAS instruments, 365 nm IBBCEAS and 460 nm IBBCEAS are used hereafter, and the 460 nm IBBCEAS have been described detailly in another paper (Ling et al., 2013).

A two-day atmospheric measurement of $NO_2$ was deployed on the sixth floor of the laboratory building (northwest of Hefei City, China) from 11:00 am, March 14th, 2014, to 11:00 am, March 16th, 2014. The two IBBCEAS instruments shared the



same sample inlet. The sample flow was filtered with a PTFE filter to remove atmospheric aerosols and then divided into two gas lines into the two instruments. Time series of $NO_2$ by the 365 nm IBBCEAS and 460 nm IBBCEAS are shown in Fig. 4(a). During this period, concentration of $NO_2$ varied between 0.8 ppb and 60 ppb with a time-averaged value of 19.5 ppb. The combined data sets are shown in Fig. 4(b) with least-square fitting giving $R^2 = 0.996$, a slope of 0.988 and an intercept of 0.50 ppb, which demonstrates the accuracy of the proposed method in deriving $d_{eff}$ for the 365 nm IBBCEAS.

### 3.3 Sample loss and secondary formation.

### 3.3.1 Sample loss

Unlike DOAS or open-path IBBCEAS, close-path IBBCEAS instrument has wall surfaces which means that potential sample loss and secondary formation for HONO on surfaces should both be considered.

To determine of the sample loss of HONO, a HONO standard generator was developed to supply stable concentrations of HONO (Kleffmann et al., 2004; Liu et al., 2016). The schematic of the HONO standard generator is shown in Fig. 5(a), and it contains a mixed solution of $NaNO_2$, dilute sulfuric acid, a peristaltic pump, a mass flowmeter, a thermostatic bath, and a custom-made spiral tunnel. The mixture solution of $NaNO_2$ and dilute sulfuric acid and the spiral tunnel were both cooled by the thermostatic bath. The mixed liquor was pumped into the spiral tunnel by a peristaltic pump, and pure nitrogen was simultaneously added as controlled by a mass flowmeter. Under acidic conditions, $HNO_2$ (l) dissolved in the solution evaporates. If temperature (5 ℃) and flow rates of the mixture solution (2 mL/min) and nitrogen (2 L/min) are held constant, a stable concentration of gaseous HONO can be generated. And we could change the HONO concentration easily by changing the key parameters of the HONO standard generator.

This experiment was operated in the laboratory as following: 1. HONO from the standard generator directly flowed into the IBBCEAS instrument, with its steady-state concentration measured by IBBCEAS. 2. The HONO source was replaced by a fast $N_2$ flow (10 SLPM) which was kept running for about 5 mins. 3. A second 1 μm PTFE filter and 3-m length PFA inlet tube and a piece of PFA tube of the same dimension as that of the optical cavity were added upstream of the IBBCEAS cavity to reproduce any potential loss on the particle filter, inlet and cavity walls. The HONO flow was re-introduced through the extra components and the IBBCEAS cavity and the new steady-state HONO concentration was measured by IBBCEAS. 4. The particle filter, PFA inlet tube, and the PFA "cavity" tube were all removed, and pure nitrogen was again flowed through the IBBCEAS instrument. In this experimental cycle, the sample loss of the IBBCEAS instrument for HONO was found to be about 2% (from average 46.0 ppb to average 45.1 ppb), as shown in Fig. 5(b).

Meanwhile, we found that once the fast $N_2$ flow was introduced into the IBBCEAS, the observed [HONO] rapidly fell below the detection limit, suggesting that the amount of HONO adsorbed on the surface of the filter or PFA tubes might be negligibly small.

In addition, varying the current supply could change the emission intensity of the LED light source. The observed [HONO] saw little change, suggesting that photolysis of HONO by the 365 nm UV-LED inside the optical cavity is negligible.




### 3.3.2 Secondary formation

To investigate any potential secondary HONO formation on the inlet or cavity walls from $NO_2$, about 80 ppb $NO_2$ (the relative humidity was about 50%) was flown through a 3-m PFA inlet tube into the IBBCEAS instrument for a long time at typical sampling flow rates. No [HONO] was observed in the optical cavity, suggesting that the secondary HONO formation is negligible for this IBBCEAS instrument under this typical operation condition. But for the complex field condition, possible contribution from artificially produced HONO along the inlet line and/or in instrument are more difficult to characterize under different atmospheric conditions. For this work, we have made efforts to minimize HONO sampling artifacts. The exclusively PFA Teflon inlet and optical cavity were used to minimize the generation and wall loss of HONO, well known for its chemical inertness, and the air flow rate was increased to reduce the residence time.

### 3.4 Instrument's stability and detection limit

An Allan variance analysis was carried out to evaluate the sensitivity and long-term stability of the IBBCEAS instrument. First, pure $N_2$ spectra was recorded using a long sequence of 10,046 measurements of $I(\lambda)$ (integration time = 3 s per spectrum. A 120 s subset of the aforementioned spectra were average to yield an $I_0(\lambda)$. Time series of "HONO" retrieved from the pure $N_2$ spectra was then retrieved using Eq.1. Second, this time series of HONO were used to generate a number of sets of time series corresponding to different averaging times, $t$, and different total number of elements at each averaging time, $N$. This used averaging of successive measurements (e.g. a time series of $N = 10,046$ measurements for $t = 3s$; a time series of $N = 5023$ measurements for $t = 6$ s, etc.; $N_{min} = 10$). Finally, the standard variance, $\sigma^2_{S_{HONO}}(t)$, and the Allan variance, $\sigma^2_{A_{HONO}}(t)$, of the time series were calculated for each integration time using the following:

$$\sigma^2_{S_{HONO}}(t) = \frac{1}{N-1} \sum_{i=1}^{N-1} (X_{HONO_i}(t) - \mu)^2 \qquad (4)$$

$$\sigma^2_{A_{HONO}}(t) = \frac{1}{2(N-1)} \sum_{i=1}^{N} (X_{HONO_{i+1}}(t) - X_{HONO_i}(t))^2 \quad (5)$$

where $X_{HONO_i}(t)$ from $i = 1$ to $N$ was the retrieved [HONO] in the time series obtained at each averaging time $t$, and $\mu$ is the average concentration of HONO over the entire period. The standard deviation $\sigma_{S_{HONO}}(t)$ provides a measure of the instrument's detection limit for a given integration time while the Allan deviation $\sigma^2_{A_{HONO}}(t)$ indicates the instrument's long-term stability.

For a measurement dominated by random white noise, the Allan deviation decreases as $\sqrt{t}$ with a gradient = -0.5 as shown in Fig. 6. At longer averaging times, the difference between successive measurements (i.e. two at adjacent ts) is comparable to the systematic time-dependent drifts in the instrument. Increasing integration time yields no benefit in decreasing the Allan



deviation if the integration time exceeds 597s. The minimum of the Allan deviation plot indicates the averaging time that produces maximum detection accuracy. To capture the rapid variation of HONO in the field, time resolution of the IBBCEAS instrument was typically set to 30 s, and the inferred 1σ detection limit of HONO is about 90 ppt at this integration time. We compared the detection limit and time resolution of different IBBCEAS instruments for HONO

measuremnt, and this home-built IBBCEAS instrument has shown reasonaly good performance, as is shown in Table 1.

## 3.5 Concentration retrieval

The HONO and $NO_2$ concentrations are retrieved via least-square fitting of their absorption cross sections to the measured absorption coefficient $\alpha(\lambda)$ using the retrieval software DOASIS (Kraus, 2006). These cross-sections were obtained by convolving the high resolution reference cross section of HONO (Stutz et al., 2000), $NO_2$ (Voigt et al., 2002) and $O_4$ (Fally et

al., 2000) with an instrument function of 0.40 nm FWHM. The reference spectrum adjustment parameters such as central wavelength shift and the peak squeeze were considered within the retrieval software and a fifth-order polynomial was used to account for smooth variation in the spectral background. To determine the optimum wavelength region for fitting, the emission profile of the LED light source, the absorption peak of the measured gases, the mirror reflectivity and the correlations for different wavelength intervals (to minimize the level of correlation, as shown in Fig. 7) were

comprehensively considered. The fitting range of HONO and $NO_2$ was finally chosen to be 359-387 nm to reduce the cross interference of these two absorbing gases.

Figure 8 presents data on HONO and $NO_2$ measurements from an absorption spectrum of ambient air in Wangdu. The concentrations of HONO and $NO_2$ were 2.53 ppb and 3.05 ppb at 12: 44 AM, June 28[th], 2014, respectively.

The total relative uncertainty of the IBBCEAS instrument was estimated to be approximately 9% considering the uncertainty

in the cross sections (5%), reflectivity (5%), spectral fitting (4%), effective cavity length (3%), pressure in the cavity (1%), $\Delta I/I_0$ (1%) and sample loss(0.5%).

## 4 Results and Discussion

### 4.1 Field inter-comparison of IBBCEAS vs. LOPAP for HONO, and IBBCEAS vs. BLC-NO₂ analyzer for NO₂

A field inter-comparison was carried out at a rural site (N38.68, E115.18; Northern China) in Wangdu, Hebei Province

between June and July, 2014 to validate the performance of the IBBCEAS instrument. This was done during the Campaigns of Air Pollution Research in Megacity Beijing and North China Plain (CAREBEIJING-NCP, 2014). The site was located in farmland area (mainly wheat and corn) with few nearby factories. The only main highway (G4) near the site was being rebuilt and blocked, and thus there was little influence from traffic. All instruments were placed in seven temperature-stabilized lab containers (Fig. 9). The IBBCEAS instrument was placed inside one that was stacked on top of another

container; hence, the sampling height of the IBBCEAS instruments was about 7 m above the ground.



The IBBCEAS instrument introduced in this paper was compared with a long optical path absorption photometer (LOPAP) that is the same HONO instrument used during the Zeppelin measurements (Li et al., 2014). Fig. 10(a) and Fig. 10(c) show that the [HONO] measured by this IBBCEAS instrument agreed well with that by LOPAP with $R^2 = 0.894$. The regression of LOPAP [HONO] against the IBBCEAS [HONO] yields a slope of 0.94 with an offset of 0.10 ppb. Note that the time resolution of the IBBCEAS instrument (< 1 min) is higher than that of the LOPAP instrument (5 min). Thus, the data were averaged to every 5 min, and some of the rapid variations of [HONO] were captured by the IBBCEAS instrument only.

In addition, $NO_2$ concentrations were compared with those from a chemiluminescence analyzer (Thermo Fisher Electron Model 42i equipped with a Blue Light Converter, which is noted as "BLC-$NO_2$ analyzer") to verify the accuracy of the IBBCEAS instrument for $NO_2$. The magnitude of $R^2$ (0.952) suggests very good agreement between the two instruments (Fig. 10(b) and Fig. 10(d)). The regression of BLC-$NO_x$ [$NO_2$] against IBBCEAS [$NO_2$] resulted in a slope of 0.96 with an offset of -0.12 ppb.

Compared to the field inter-comparison between IBBCEAS and LOPAP in Hong Kong(Wu et al., 2014), this agreements demonstrated that our IBBCEAS instrument can achieve robust performance and system stability in the field and the IBBCEAS technique offers promising prospect for measuring environmental HONO and $NO_2$ at their typical ambient levels.

## 4.2 Mobile measurements of HONO and NO₂

Due to its compact design, the IBBCEAS instrument can be easily deployed in a car to measure the spatial distribution of HONO and $NO_2$. In a trial, the IBBCEAS instrument was powered by the lithium battery, and the ambient air was drawn into the IBBCEAS instrument though an inlet fixed on the roof of the car. The route of the experiment was around the Beijing International Airport to measure the daytime traffic emission in Beijing from 11:00-13:00 on Dec 10[th], 2014. The day of the measurement was sunny and the average temperature was about 4 ˚C. Because of the prevailing strong Northern wind, the air mass was relatively clean with a mean $PM_{2.5}$ concentration of about 14 μg/m³. The spatial distributions of HONO and $NO_2$ from the 30 s averaged data were shown in Fig. 11, with the HONO and $NO_2$ concentrations ranging from the instrument detection limit to 2.0 ppb and 49.6 ppb, respectively, and the mean concentrations of HONO and $NO_2$ being 0.25 ppb and 16.82 ppb, respectively. The high values of HONO and $NO_2$ were both mainly found at the crossroads with heavy traffic. At the maximum HONO concentration captured en route (~2.0 ppb), [HONO] / [$NO_2$] ratio was about 4.1 %, and the location where this maxima HONO concentration occurred was the intersection of the Jingshen Road (G101) and Huosha Road (XD20), both of which are major traffic arteries with many heavy trucks, buses and cars. We suspect the high value of HONO may have come from direct traffic emissions. Thus, the mobile IBBCEAS instrument offers a good way to study, e.g. the direct emission of HONO from onroad traffics owing to its easy deployment as well as high temporal resolution.



## 5 Conclusions

We report a home-built IBBCEAS instrument for simultaneous monitoring of environmental HONO and NO$_2$. This IBBCEAS instrument has made significant improvements on efficient sampling, vibration resistance and temperature resistance by applied of the purge flows, bypass flow and different thermostats. And the IBBCEAS instrument also has low-power consumption and a compact design that can be easily deployed on the different platforms and powered only by a high-capacity lithium ion battery.

The effective cavity length of the IBBCEAS was determined using the absorption of O$_2$-O$_2$ to account for the "shortening" effect caused by the mirror purge flows. The wall loss for HONO was estimated to 2 % via a HONO standard generator. The instrument performance was evaluated and validated, and this demonstrated the precision needed for field measurements of HONO and NO$_2$, i.e., 180 and 340 ppt in 30 s, respectively. The calculated accuracy for these measurements is about 9.0%.

The IBBCEAS instrument was verified as a useful method for measurements of HONO and NO$_2$ in polluted field environments in China. Inter-comparisons with other techniques generally showed good agreement, demonstrating the promising potential of using IBBCEAS for sensitive, accurate and fast simultaneous measurements of HONO and NO$_2$ in the future.

*Acknowledgements.* This work was supported by the National Natural Science Foundation of China (Grant No. 91544104, 41571130023, 41705015 and 21190052).

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



**Table 1.** Characteristics of typical IBBCEAS instruments for HONO measurements

| Research Unit | Time Resolution | Detection Limit (2σ) | Field Applications | Purge flows | Reference |
|---|---|---|---|---|---|
| University College Cork | 20 s | 4 ppb | No | No | (Gherman et al., 2008) |
| Indiana University | 600 s | 0.6 ppb | No | Yes | (Scharko et al., 2014) |
| Nanchang Hangkong University Université du Littoral Côte d'Opale | 120 s | 0.6 ppb | Hong Kong | Yes | (Wu et al., 2014) |
| University of Colorado NOAA | 5 s | 0.35 ppb | US & China | No | (Min et al., 2016) |
| Tokyo Agriculture and Technology | 600 s | 0.4 ppb | Japan | Yes | (Nakashima and Sadanaga, 2017) |
| Anhui Institute of Optics and Fine Mechanics, CAS | 30 s | 0.18 ppb | China | Yes | This work |

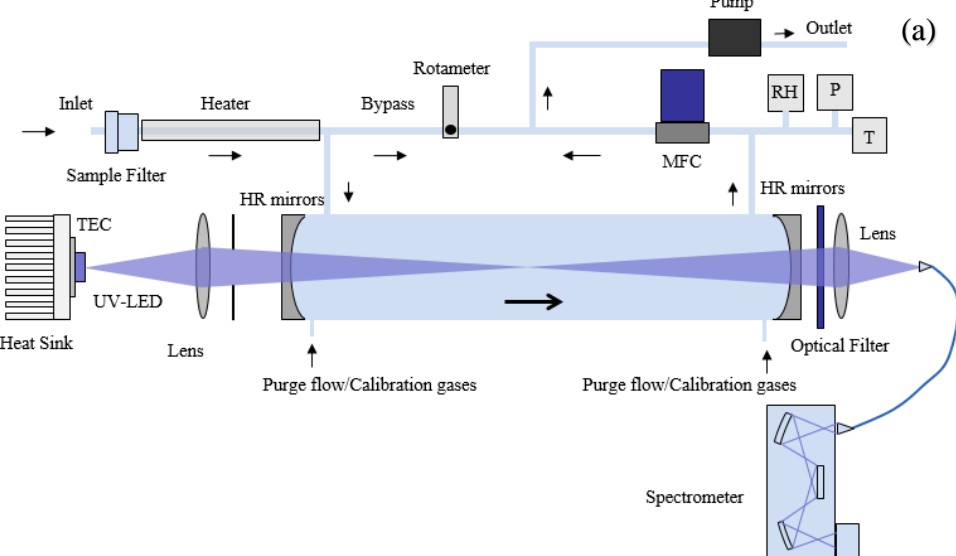

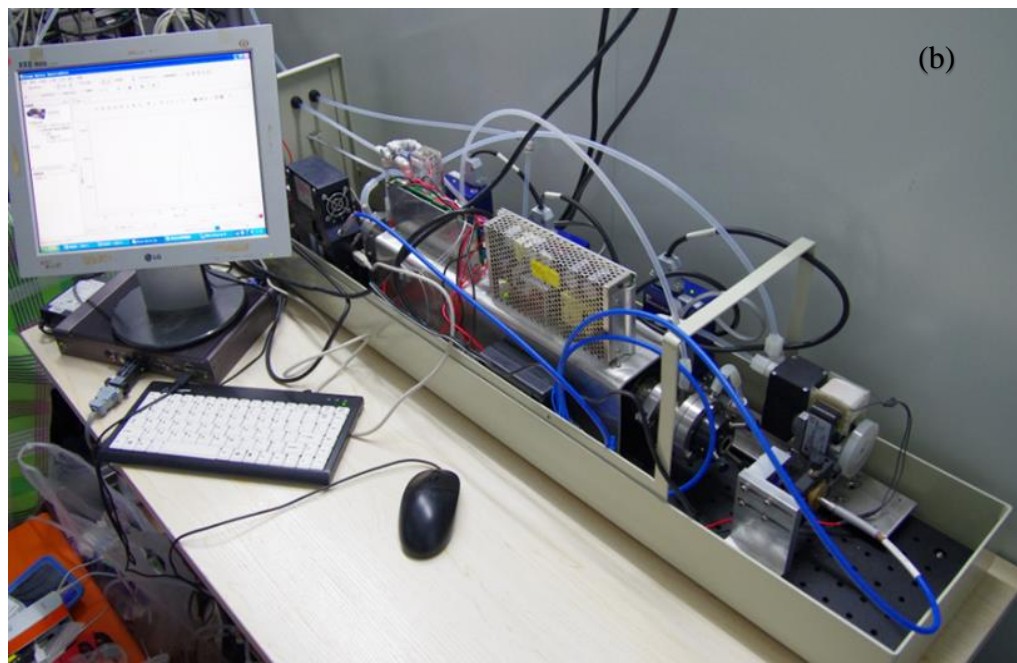

**Fig. 1. (a)** Schematic of the IBBCEAS instrument; **(b)** Photograph of the IBBCEAS instrument (the case was removed for this picture to show the structure of the IBBCEAS instrument).



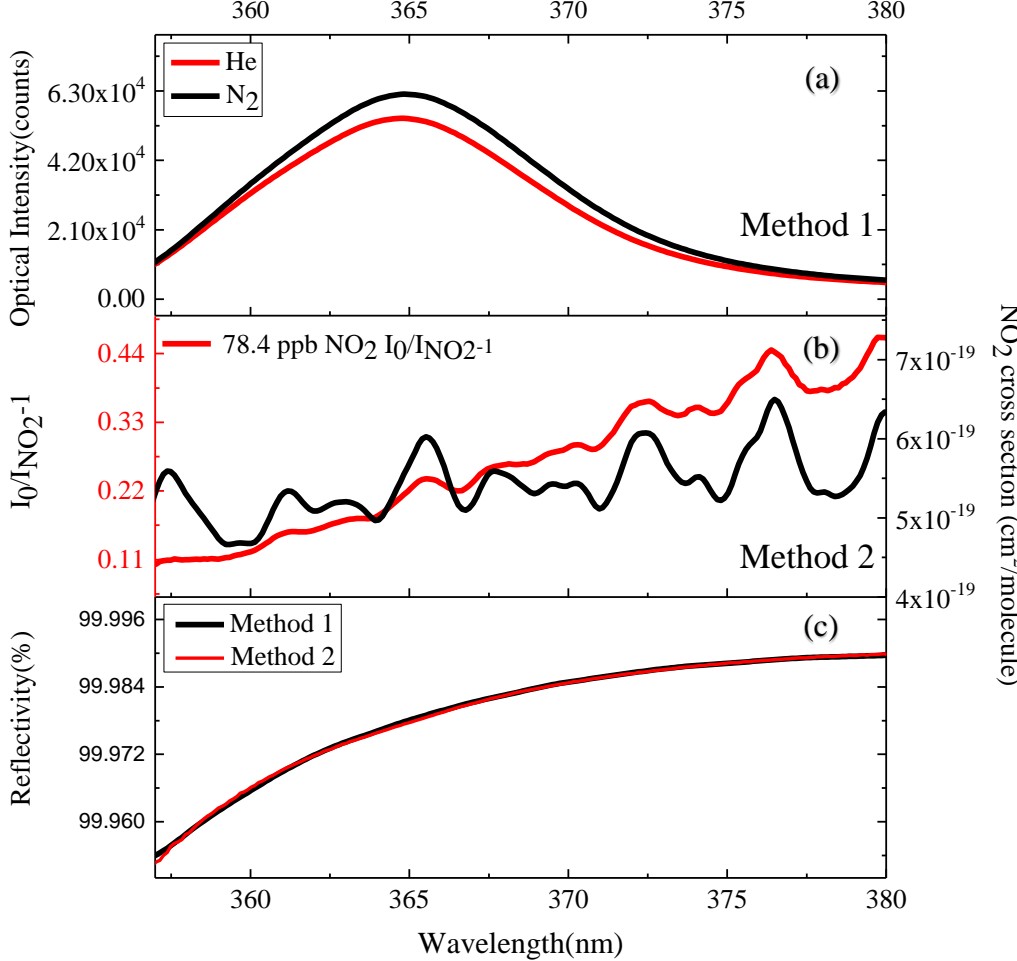

**Fig. 2. (a)** The red curve is the transmitted intensity spectrum when the cavity is filled with He, and the black curve is that when the cavity is filled with $N_2$; **(b)** The red curve is the absorption spectrum of 78.4 ppb $NO_2$ seen by IBBCEAS, and the black curve is the $NO_2$ cross section; **(c)** The derived curve of mirror reflectivity.




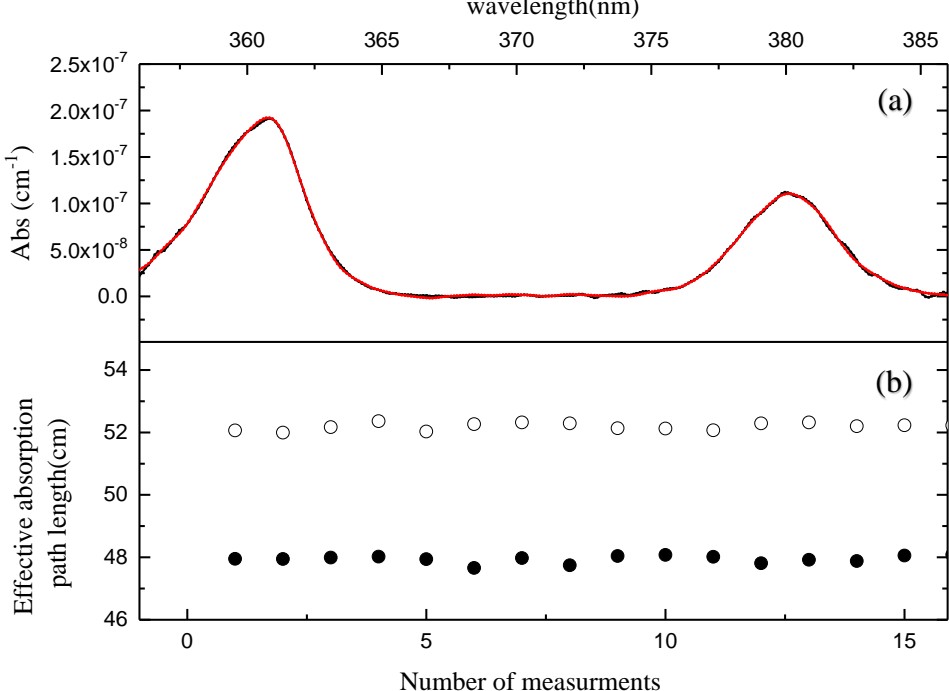

**Fig. 3. (a)** Examples of concentration retrieval with $O_2$-$O_2$. Here, the black line is the measured spectrum and the red line is the fitted $O_2$-$O_2$ absorption spectrum. **(b)** The black dots are the effective absorption path length with 0.1 SLPM purge flows and 1 SLPM sampling flow, and the circles are the effective absorption path length with 0.1 SLPM purge flows and 3 SLPM sampling flow.





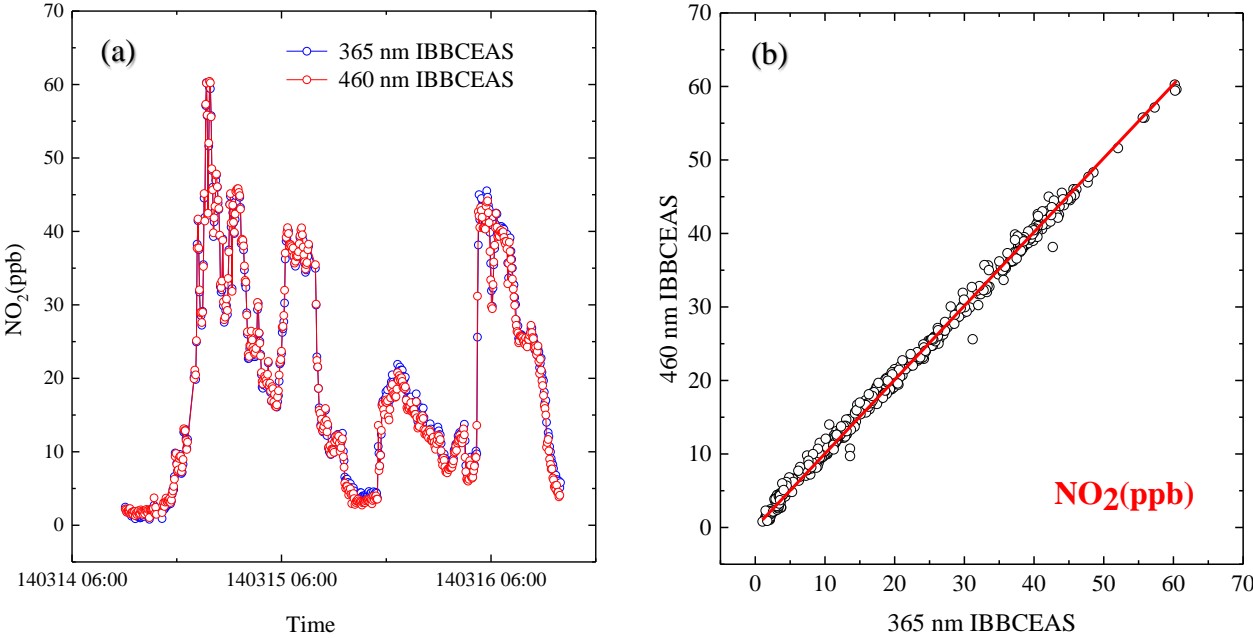

**Fig. 4. (a)** Time series measurements of NO2 from the 365 nm IBBCEAS and 460 nm IBBCEAS. **(b)** Correlation plot of between the two NO$_2$ measurements.



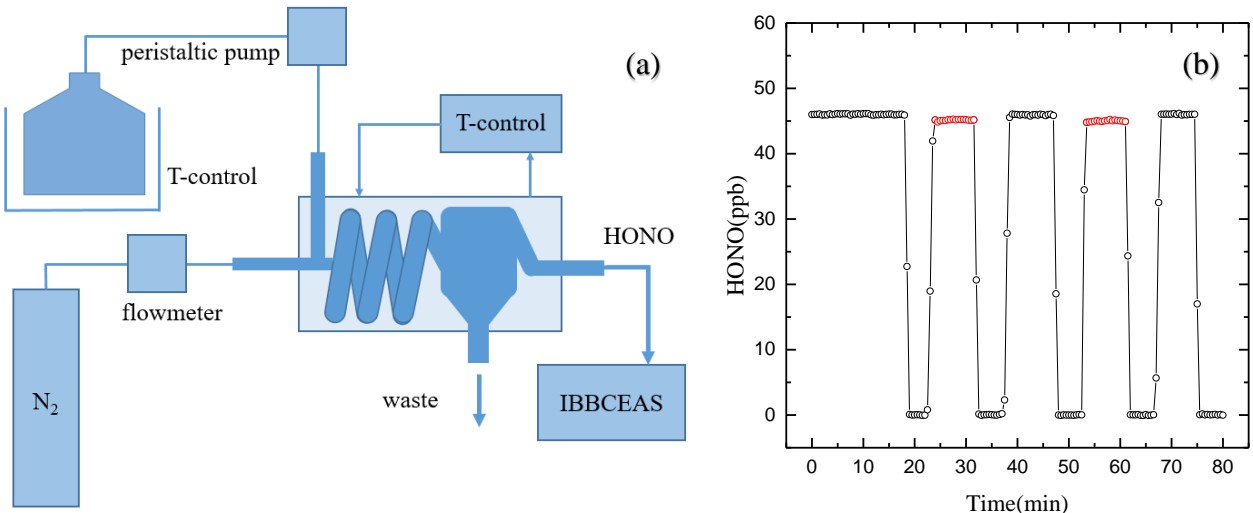

**Fig. 5. (a)** The schematic of HONO standard generator. **(b)** Black circles correspond to observed [HONO] from the HONO standard generator, Red circles correspond to the measured [HONO] with the extra 1 m PTFE filter, 3-m PFA inlet tube and the simulative PFA optical cavity tube added in front of the IBBCEAS instrument.



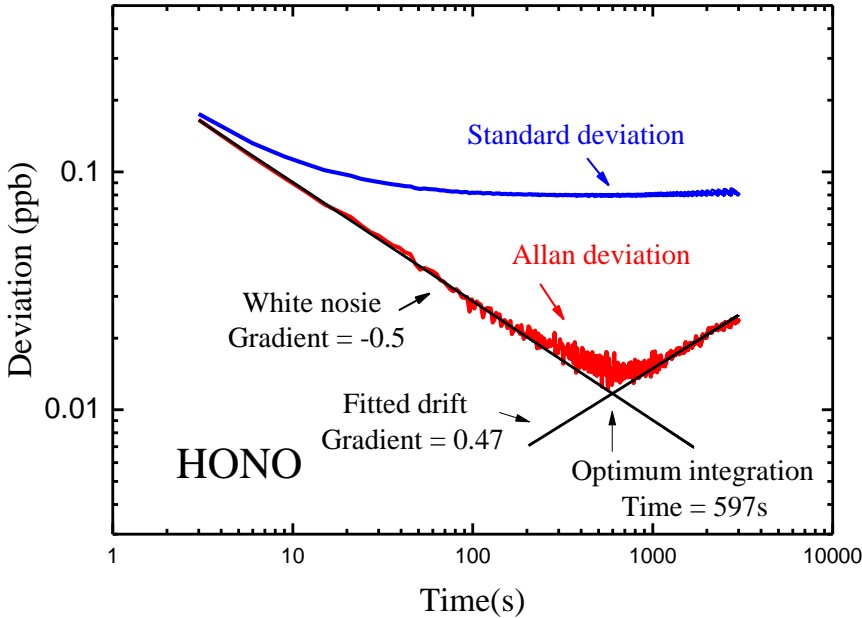

**Fig. 6.** Standard deviation and Allan deviation plots for measurements of HONO. For integration times of below 110 s the Allan deviation decreases approximately as $\sqrt{t}$ (gradient of -0.5). The minima of the Allan deviation indicate the optimum averaging time for maximum detection performance.




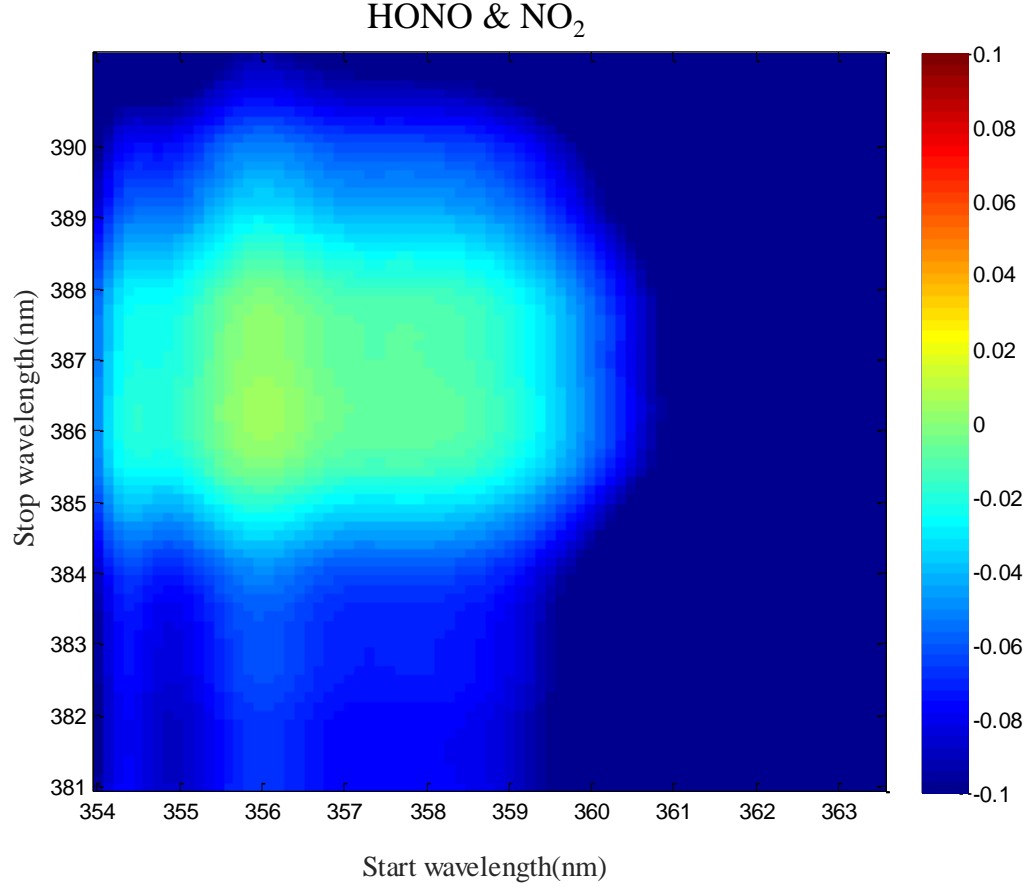

**Fig. 7.** Overall correlation (expressed as the root mean square of the non-diagonal elements of the correlation matrix) for different wavelength intervals in the 354 nm - 390 nm wavelength range.



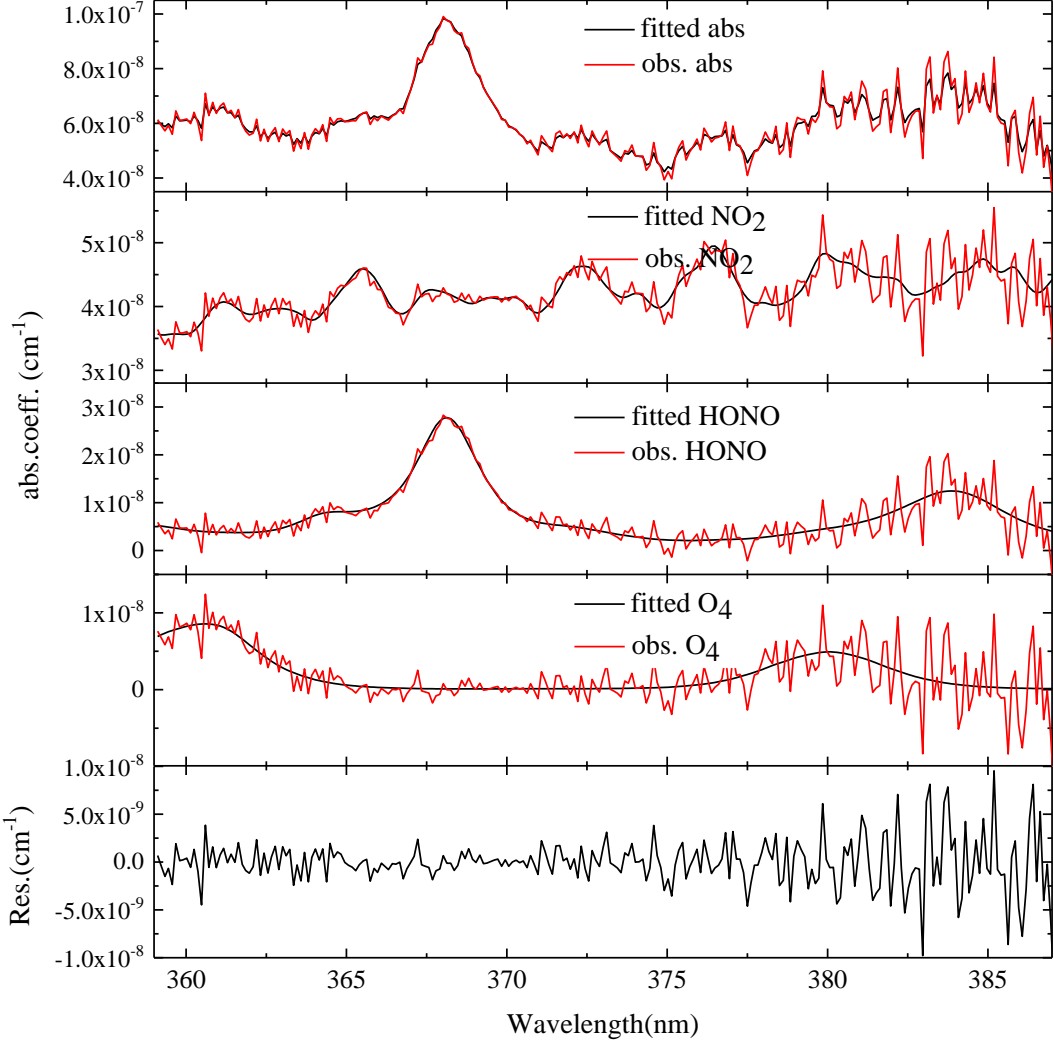

**Fig. 8.** Example of retrieved and fitted absorption spectra of $NO_2$ and HONO measured at 12:44 AM on June 28th. The top panel shows the absorption measurement by IBBCEAS. The absorption features due to $NO_2$ and HONO can be clearly seen. The second, third and fourth panels show the same spectrum plotted in the first panel. They are decomposed into individual absorption contributions from $NO_2$ (3.05 ± 0.17 ppb), HONO (2.53 ± 0.093 ppb), and $O_4$ ([$O_2$] is close to 21 %). The fifth panel is the residual spectrum (standard deviation = 2.76405×$10^{-9}$ $cm^{-1}$).



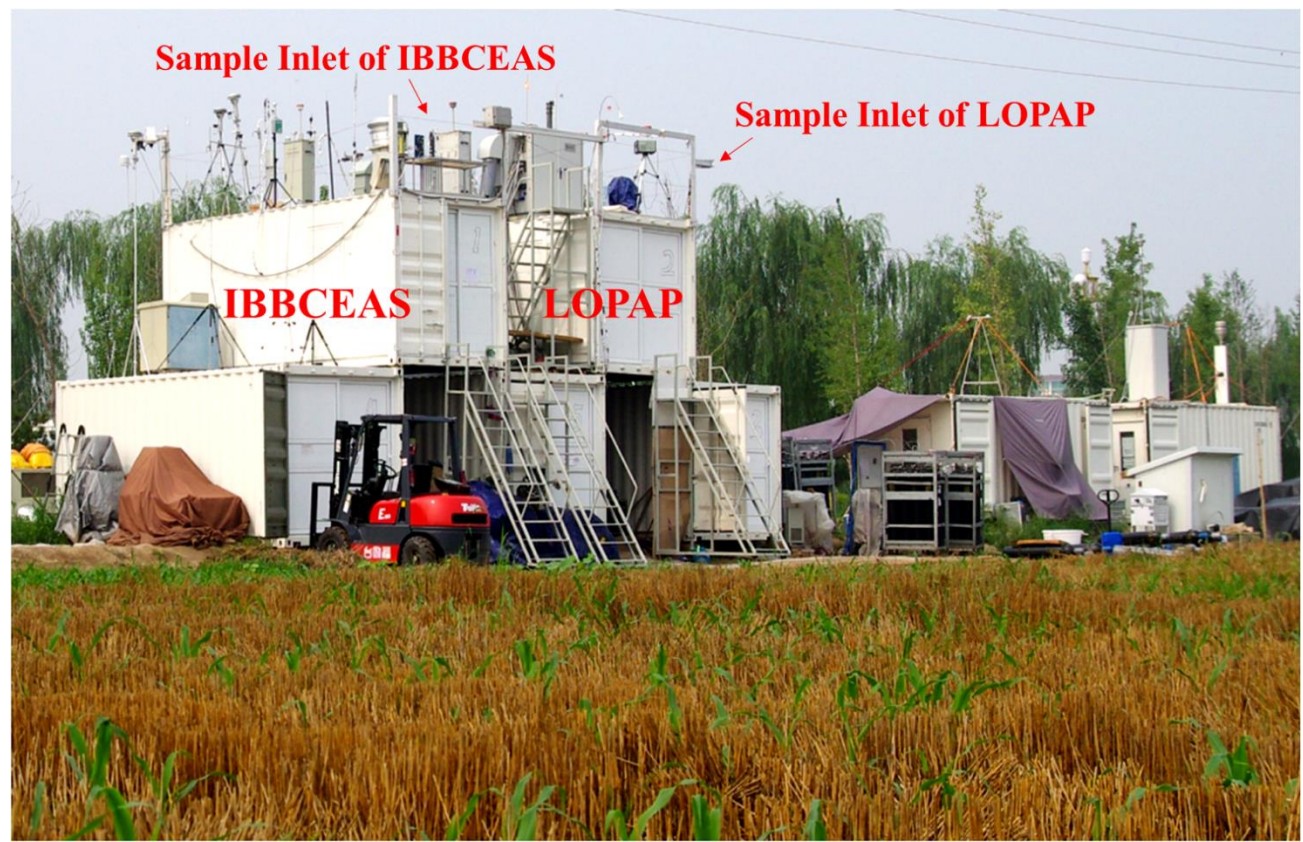

**Fig. 9.** The measurement site of CAREBEIJING and the container in which the IBBCEAS instrument was placed.





**Fig. 10. (a)** Time series of [HONO] measured by the IBBCEAS instrument and LOPAP; **(b)** Time series of [NO₂] measured

5  by the IBBCEAS instrument and the BLC-NOₓ analyzer; **(c)** Correlation plot of the two [HONO] time series (data averaged
to 5 min base); **(d)** Correlation plot of the two [NO₂] time series (data averaged to 5 min base).



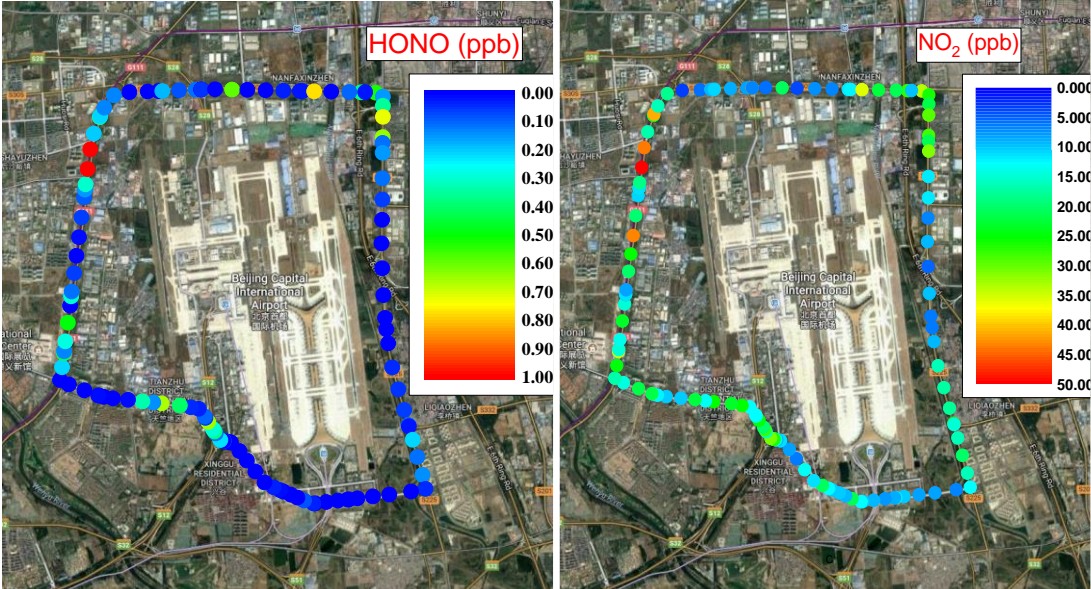

**Fig. 11.** Results of the HONO and NO₂ around the Beijing International Airport.