# Peer review of "Development of an incoherent broadband cavity enhanced absorption spectrometer for in situ measurements of HONO and NO2"

_Atmospheric Measurement Techniques, 2017_

## Referee Comment (RC1) · Anonymous Referee #1 · 27 Mar 2018

General comments: This paper describes details of a newly developed HONO and $NO_2$ simultaneous measurement system based on an incoherent broadband cavity enhanced absorption spectrometer (IBBCEAS). There have been recent several reports of a HONO and $NO_2$ simultaneous measurement system so that measurement principle reported in this article is not novel. But I think IBBCEAS is revolutionary measurement system of HONO and evolution of the HONO measurement system is important in order to clarify behaviors (i.e. source, sink, reactions in the atmosphere, and so on) of HONO. In this paper, some performances are improved (e.g. stability, detection limit, and so on) and the authors developed the mobile measurement system of HONO and $NO_2$. I recommend this paper to be published in Atmospheric Measurement Tech-

niques. However, I found several dubious points in this paper. The authors should revise appropriately.

Specific comments:

Title: I think "in China" can be deleted. I think this system can measure HONO and $NO_2$ simultaneously in other country as well as China.

Page 2, lines 3-4 "such as $O_3$": $O_3$ is one of photochemical smog, so that other secondary pollutants are recommended (e.g. $HNO_3$).

Page 8, lines 4-5 "a slope of 0.988 and an intercept of 0.50 ppb,": The authors should add errors of a slope and an intercept.

Section 3.3.1: For HONO generation, do the authors confirm simultaneous generation of $NO_2$?

Section 3.3.1: The authors should state relative humidity (RH) as an experimental condition, and should discuss an RH dependence of the HONO loss.

Section 3.3.1: $NO_2$ loss is negligible? The authors should comment the $NO_2$ loss as well as the HONO loss.

Section 3.3.2: The authors should discuss an RH dependence of the secondary HONO formation.

Page 11, line 4 "a slope of 0.94 and an intercept of 0.10 ppb,": The authors should add errors of a slope and an intercept.

Page 11, lines 10-11 "a slope of 0.96 and an intercept of –0.12 ppb,": The authors should add errors of a slope and an intercept.

Technical corrections:

Page 1, line 29: NOx → $NO_x$

Page 1, line 30: $R^2$ → $R^2$

Page 2, line 11: (Jr et al., 1984; → (Pitts et al., 1984;

Page 2, line 29: (J et al., 2001) → (Heland et al., 2001)

Page 4, line 11: f = 60 mm → $f$ = 60 mm

Page 13, line 10: J, H., J, K., R. K., and P, W.: → Heland, J., Kleffmann, J., Kurtenbach, R., and Wiesen, P.:

Page 13, line 15: Jr, J. N. P., → Pitts, J. N., Jr.,

Page 15, line 30: NO2 → $NO_2$
* * *

---

## Referee Comment (RC2) · Anonymous Referee #3 · 2 May 2018

The paper describes an optical instrument for measuring concentration of HONO and NO2 in the atmosphere. The instrument can be put on a mobile platform and results using such a platform are presented. I think that the main shortcoming of the paper is that the authors do not make sufficiently clear what it is that is new about their instrument. Is it the choice of wavelength that enables O2-O2 to be used as a gas to measure the effective cavity length? Or is it a series of incremental changes that allow it to be used outdors in dirty, vibrating or temperature varying environments? Other instruments of the same kind are described, but little that is specific is said about their shortcomings, and how this instrument overcomes these issues.

[Figure]

Some more specific issues follow: I agree with the other referee that this is not an instrumrnt just for China - surely it is for the world?

The authors should delete the words "home made" - this is not very positive, and the fact that they have made the instrument is enough.

In the abstract, sentence 3, some rewording is in order. Instruments don't make significant improvements, their makers do.

One of the references is incomplete (J et al.) and the reference to Krauss is quite insufficient for a reader to follow up.

In section 2.1 the Beer-Lambert law ought to be quoted so as to properly introduce extinction. There is also something odd about the sentence order, where sentence 3 seems out of place in between two others on extinction.

Similarly in section 2.3 the order in which the concepts are presented seems odd. It would be improved if the two paragraphs were swapped, but it could probably do with more work.

the beginning of section 2.4 is a bit ambiguous - it could be construed that the current IBBCEAS is not the improved one, but that the improved one is a different one. Also, what is "temperature resistance"? Is it the reciprocal of thermal conductivity? Sentence 2 of this section is vague; what changes? How much?

The lines after equation 2 are hard to follow as there is an incomplete sentence - I think possibly a comma has been replaced with a full stop?

In section 3.2.2 I think there should be a reference for these absorption peaks. When you refer to particle free gasses here it suggests that there are gasses other than O2 used. Also you have not in fact shown that the purge flow is significant because you haven't given the actual physical length of the cavity.

In section 3.2.3, "detailly" is not a word in English

---

## Author Comment (AC1) · 29 May 2018

**Reviewer # 1**

**Comments and suggestions:**

1. Title: I think "in China" can be deleted. I think this system can measure HONO and NO2 simultaneously in other country as well as China.

**Response**

Thank you for your suggestion, we deleted "in China" in the title.
* * *
**Comments and suggestions:**

2. Page 2, lines 3-4 "such as O3": O3 is one of photochemical smog, so that other secondary pollutants are recommended (e.g. HNO3).

**Response**

Yes, we agree and we have made change as suggested. We added "has been found worldwide to play a key role in enhancing the formation of photochemical smog and other secondary pollutants such as $O_3$, $HNO_3$ and secondary organic aerosols in polluted atmospheric boundary layers Please see P2. Line 4.
* * *
**Comments and suggestions:**

3. Page 8, lines 4-5 "a slope of 0.988 and an intercept of 0.50 ppb,": The authors should add errors of a slope and an intercept.

**Response**

Thank you for your suggestion. It was corrected to: "a slope of $0.988 \pm 0.0027$ and an intercept of $0.503 \pm 0.0064$ ppb." Please see P8 Line 6 - 7.
* * *
**Comments and suggestions:**

4. Section 3.3.1: For HONO generation, do the authors confirm simultaneous generation of NO2?

**Response**

For HONO generation, no [$NO_2$] was observed by this IBBCEAS.
* * *
**Comments and suggestions:**

5. Section 3.3.1: The authors should state relative humidity (RH) as an experimental condition, and should discuss an RH dependence of the HONO loss.

**Response**

We have made change as suggested. We added "In this experimental cycle, the relative humidity(RH) was about 65 % and temperature was about 23 ℃, the sample loss of the IBBCEAS instrument for HONO was found to be about 2.0 % (from average 46.0 ppb to average 45.1 ppb), as shown in Fig. 5(b). We also repeated this experiment at different RH levels, and found that the sample loss of the IBBCEAS instrument for HONO was about 2.1 % at 25 % RH and about 1.9 % at 50 % RH, suggesting a weak RH dependence of the sample loss of the IBBCEAS instrument for HONO." Please see P8. Line 29 – P9. Line 1.
* * *
**Comments and suggestions:**

6. Section 3.3.1: $NO_2$ loss is negligible? The authors should comment the $NO_2$ loss as well as the HONO loss.

**Response**

Yes, $NO_2$ loss is negligible. We added "Furthermore, we also studied the sample loss of the IBBCEAS instrument for $NO_2$ as the similar approach and found that it was negligible." Please see P9. Line 1 - 2.
* * *
**Comments and suggestions:**

7.   Section 3.3.2: The authors should discuss an RH dependence of the secondary HONO

formation.

**Response**

We have made change as suggested. We added "To investigate any potential secondary HONO

formation on the inlet or cavity walls from $NO_2$, about 80 ppb $NO_2$ at different RH levels (about

20% RH, 30% RH, 50% RH and 70% RH) was flown through a 3-m PFA inlet tube into the

IBBCEAS instrument for a long time at typical sampling flow rates, respectively, no [HONO] was

observed in the optical cavity". Please see P9. Line 10 - 11.
* * *
**Comments and suggestions:**

Page 11, line 4 "a slope of 0.94 and an intercept of 0.10 ppb,": The authors should add

errors of a slope and an intercept.

**Response**

Thank you for your suggestion. It was corrected to: "The regression of LOPAP [HONO] against the

IBBCEAS [HONO] yields a slope of $0.941 \pm 0.0069$ with an offset of $0.110 \pm 0.0089$ ppb." Please

see P11 Line 12-13.
* * *
**Comments and suggestions:**

Page 11, lines 10-11 "The regression of BLC-$NO_x$ [$NO_2$] against IBBCEAS [$NO_2$] resulted in

a slope of 0.964 with an offset of -0.123 ppb": The authors should add errors of a slope and an

intercept.

**Response**

Thank you for your suggestion. It was corrected to: "The regression of BLC-$NO_x$ [$NO_2$] against

IBBCEAS [$NO_2$] resulted in a slope of 0.964 ± 0.0042 with an offset of -0.123 ± 0.0539 ppb."

Please see P11 Line 20-21.
* * *
**Comments and suggestions:**

Page 1, line 29: NOx → $NO_x$

Page 1, line 30: R² → $R^2$

Page 2, line 11: (Jr et al., 1984) → (Pitts et al., 1984)

Page 2, line 29: (J et al., 2001) → (Heland et al., 2001)

Page 4, line 11: f = 60 mm → $f$ = 60 mm

Page 13, line 10: J, H., J, K., R. K., and P, W.: → Heland, J., Kleffmann, J., Kurtenbach,

R., and Wiesen, P.:

Page 13, line 15: Jr, J. N. P., → Pitts, J. N., Jr.,

Page 15, line 30: NO2 → $NO_2$

**Response**

Corrected.

---

## Author Comment (AC2) · 29 May 2018

**Reviewer # 3**

**Comments and suggestions:**

1. I think that the main shortcoming of the paper is that the authors do not make sufficiently clear what it is that is new about their instrument. Is it the choice of wavelength that enables O2-O2 to be used as a gas to measure the effective cavity length? Or is it a series of incremental changes that allow it to be used outdors in dirty, vibrating or temperature varying environments? Other instruments of the same kind are described, but little that is specific is said about their shortcomings, and how this instrument overcomes these issues.

**Response**

This paper describes details of a newly developed HONO and $NO_2$ simultaneous measurement system with significant improvements in efficient sampling, vibration resistance and temperature resistance by applied of the purge flows, bypass flow and different thermostats. And as our knowledge, this is the first application of determining the effective cavity length by pure oxygen, and we discussed the wall loss of for HONO and $NO_2$, HONO adsorbed on the surface, photolysis of HONO by the 365 nm UV-LED light source and secondary HONO formation based on IBBCEAS technique in detail. We think they are the innovation points about our instrument.
* * *
**Comments and suggestions:**

2.  I agree with the other referee that this is not an instrument just for China - surely it is for the world?

**Response**

Yes, we deleted "in China" in the title.
* * *
**Comments and suggestions:**

3.  The authors should delete the words "home made" - this is not very positive, and the fact that

they have made the instrument is enough.

**Response**

Thank you for your suggestion, we deleted the words "home made".
* * *
**Comments and suggestions:**

4.  In the abstract, sentence 3, some rewording is in order. Instruments don't make significant improvements, their makers do.

**Response**

Thank you for your suggestion, it was corrected to: "To achieve robust performance and system stability under different environment conditions, the current IBBCEAS instrument has been developed with significant improvements in terms of efficient sampling as well as resistance against vibration and temperature change." Please see P1. Line 21-23.
* * *
**Comments and suggestions:**

5.  One of the references is incomplete (J et al.) and the reference to Krauss is quite insufficient for a reader to follow up.

**Response**

Corrected.
* * *
**Comments and suggestions:**

6.  In section 2.1 the Beer-Lambert law ought to be quoted so as to properly introduce extinction. There is also something odd about the sentence order, where sentence 3 seems out of place in between two others on extinction.

Similarly in section 2.3 the order in which the concepts are presented seems odd. It would be improved if the two paragraphs were swapped, but it could probably do with more work.

**Response**

Thank you for your suggestion, the order of the sentences in section 2.1 and section 2.3 have been modified and the two paragraphs were swapped. Please see P3 Line 28 – 31 and P4 Line 23 – P5 Line 9.
* * *
**Comments and suggestions:**

7. The beginning of section 2.4 is a bit ambiguous - it could be construed that the current IBBCEAS is not the improved one, but that the improved one is a different one. Also, what is "temperature resistance"? Is it the reciprocal of thermal conductivity? Sentence 2 of this section is vague; what changes? How much?

**Response**

Thank you for your suggestion, it was corrected to: "This IBBCEAS instrument has been developed with significant improvements in terms of efficient sampling as well as resistance against vibration and temperature change to achieve robust performance in different field environments. For example, we found that the parameters of the CCD spectrometer changed with temperature especially when it dropped below 5 °C. To ensure the IBBCEAS can work across a wide ambient temperature range (5 ~ 35 °C)." Please see P5 Line 12 – 18.
* * *
**Comments and suggestions:**

8. The lines after equation 2 are hard to follow as there is an incomplete sentence - I think possibly a comma has been replaced with a full stop?

**Response**

Corrected. Please see P6 Line 10 – 11.
* * *
**Comments and suggestions:**

9.  In section 3.2.2 I think there should be a reference for these absorption peaks. When you refer to particle free gasses here it suggests that there are gasses other than $O_2$ used. Also you have not in fact shown that the purge flow is significant because you haven't given the actual physical length of the cavity.

**Response**

Thank you for your suggestion, we added" In the spectral range of the 365 nm LED, $NO_2$(Voigt et al., 2002), HONO(Stutz et al., 2000) and $O_2$-$O_2$ collision pair(Greenblatt et al., 1990) have clear absorption peaks. " Please see P7 Line 12 - 13, and we added" Using this method, the geometric cavity length of the IBBCEAS instrument was 55.0 cm and the effective cavity length was determined to be $48.0 \pm 0.5$ cm."Please see P7 20 - 23.
* * *
**Comments and suggestions:**

10.  In section 3.2.3, "detailly" is not a word in English.

**Response**

Corrected.